# T Regulatory CD4^+^CD25^+^FoxP3^+^ Lymphocytes in the Peripheral Blood of Left-Sided Colorectal Cancer Patients

**DOI:** 10.3390/medicina55060307

**Published:** 2019-06-25

**Authors:** Katarzyna Dylag-Trojanowska, Joanna Rogala, Radoslaw Pach, Maciej Siedlar, Jaroslaw Baran, Marek Sierzega, Justyna Zybaczynska, Marzena Lenart, Magdalena Rutkowska-Zapala, Antoni M. Szczepanik

**Affiliations:** 1Department of General and Oncological Surgery, John Gawlik Hospital, 22 Szpitalna Street, 34-200 Sucha Beskidzka, Poland; kadgta@interia.pl; 2First Department of General, Oncological and Gastroenterological Surgery, Jagiellonian University Medical College, 40 Kopernika Street, 31-501 Krakow, Poland; jrogala@su.krakow.pl (J.R.); radoslaw.pach@uj.edu.pl (R.P.); marek.sierzega@uj.edu.pl (M.S.); 3Department of Clinical Immunology, Institute of Pediatrics, Jagiellonian University Medical College, 265 Wielicka Street, 30-663 Krakow, Poland; misiedla@cyf-kr.edu.pl (M.S.); mibaran@cyf-kr.edu.pl (J.B.); m.lenart@uj.edu.pl (M.L.); magdalena.rutkowska@uj.edu.pl (M.R.-Z.); 4Jagiellonian University Medical College, 31-008 Cracow, Poland; justyna.zybaczynska@gmail.com

**Keywords:** left-sided colon cancer, T-regulatory cells (Treg), peripheral blood, CD4^+^CD25^+^Fox3^+^ cells, prognosis, overall survival, prognostic biomarker

## Abstract

*Background and objectives:* T regulatory lymphocytes (Treg) are one of the subsets of T-lymphocytes involved in the interaction of neoplastic tumors and the host immune system, and they may impair the immune reaction against cancer. It has been shown that Treg are increased in the peripheral blood of patients with various cancers. In colorectal cancer, the prognostic role of Treg remains controversial. Colorectal cancer is a heterogenous disease, with many variations stemming from its primary tumor location. The aim of this study is to analyse the relationship between the amount of Treg in the peripheral blood of patients with left-sided colorectal cancer in various stages of disease and long-term survival. *Materials and Methods:* A prospective analysis of 94 patients with left-sided colorectal cancer and a group of 21 healthy volunteers was carried out. Treg levels in peripheral blood were analysed using flow cytometry. *Results:* There was a statistically significant difference between the amount of Treg in the Ist and IInd TNM stages (*p* = 0.047). The number of Treg in the entire study group was significantly lower than in the control group (*p* = 0.008) and between patients in stages II and III and the control group (*p* = 0.003 and *p* = 0.018). The group of pT3+pT4 patients also had significantly lower Treg counts in their peripheral blood than the control group (*p* = 0.005). In the entire study group, the level of Treg cells in the peripheral blood had no influence on survival. The analysis of the TNM stage subgroups also showed no difference in survival between patients with “low” and “high” Treg counts. *Conclusion*: The absolute number of Treg in the peripheral blood of patients with left-sided colorectal cancer was significantly decreased in comparison to healthy controls, especially for patients with stage II+III disease. Treg presence in the peripheral blood had no impact on survival.

## 1. Introduction

T-regulatory lymphocytes (Treg) are one of the subsets of T-lymphocytes involved in the interaction of neoplastic tumors and the host immune system. These cells effectively suppress the immune response against self-antigens, preventing autoimmune-reactions. In-spite of that, multiple preclinical and clinical studies have suggested that Treg cells may impair immune surveillance against cancer during oncogenesis, prevent the development of effective antitumor immunity in patients with established tumors, and possibly promote tumor progression.

However, their role in various cancers differs. In some studies, the presence of Tregs in peritumoral infiltrate was found to be a negative prognostic factor, such as in ovarian [1], breast [2], gastric [3], and pancreatic cancers [4]. On the other hand, the presence of the same cells within the tumor infiltrate was connected with better prognosis in oropharyngeal [5] and esophageal cancers [6]. In colorectal cancer, the prognostic role of Treg remains controversial [7]; some studies have reported a positive prognostic role of Treg, while others, a negative influence. T regulatory lymphocytes have also been investigated in the peripheral blood of cancer patients. Increases of Treg amounts were observed in some tumor types and sites, alongside decreases in comparison to healthy controls [8,9].

Since the composition of peritumoral infiltrate may be assessed only after surgical excision, the level of Treg cells in the peripheral blood is an attractive potential marker for repeated and long-term cancer monitoring. Examples of Treg cells serving as markers include the results of a pancreatic cancer study where peripheral blood Treg level predicted the tumor response to chemotherapy [10], and another study, where Treg levels changed markedly in rectal cancer patients after preoperative radiotherapy [11].

Colorectal cancer is one of the most frequent malignancies worldwide. According to the UICC data published in 2018, it is the third cancer in incidence and second in cancer-related mortality. Despite progress in prevention and therapy, there is still space for new therapies and translational research in colorectal cancer. In recent years, several trials on immunotherapy using checkpoint inhibitors in colorectal cancer have produced promising results [12]. Treg are involved in the mechanisms of immunosuppression and immunotolerance, which are the targets of such immunotherapy [13]. Therefore, the investigation of these cells in various subgroups of colorectal cancer patients is justified. Colorectal cancer is not a homogenous disease in terms of primary tumor location, and there is evidence that right-sided and left-sided cancers may have different biologies and prognoses [14,15,16,17]. In this study, we concentrated on tumors located in the left colon and rectum.

The aim of this study was to investigate the absolute count of T regulatory lymphocytes in the peripheral blood of patients with left-sided colorectal cancer in various stages, and its prognostic significance.

## 2. Materials and Methods

The study group consisted of 94 colorectal cancer patients treated in a single institution between 2007 and 2012. Only patients with tumors located in the rectum or left colon were included. The term left colon was defined as the large intestine, from the left 1/3 of the transverse colon distally. All patients had histologically confirmed disease, were over 18 years old, and had an electively performed surgical procedure. Patients with synchronous right-sided colon cancer or patients with a history of other neoplastic diseases were excluded. Preoperative radiotherapy was used in four of the rectal cancer patients, and two rectal cancer patients received preoperative chemoradiotherapy. All patients had no history of autoimmune diseases or recent infections. The group was composed of 39 women and 55 men, with a mean age of 65.6 (SD 9.8), in varying stages of disease (Table 1).

The surgical procedures were carried out according to oncological guidelines. Due to the changes of the TNM staging systems during the study period, all the specimens were re-staged according to the 7th edition of the TNM. The clinical and pathological data were recorded. Patients received postoperative chemotherapy if indicated. All patients were followed up for at least 5 years, or until death, and dates of death were verified by the census registry office.

The control group consisted of 21 healthy adult volunteers—9 women and 12 men—with the mean age of 44 (range from 25–55 years).

All patients provided their informed, written consent. The study was approved by the Jagiellonian University Ethical Committee KBET no 86/B/2007 and KBET no 122.6120.128.2015. The study was registered at ClinicalTrials.gov, registration number NCT03640572.

Blood samples were collected prior to any interventional procedure in sterile EDTA vacutainers. Cells preparation was started 1–2 h after a blood draw. Peripheral blood samples (100 μL) obtained from cancer patients were incubated in TruCount tubes (BD Biosciences, San Jose, CA, USA) with a monoclonal antibody cocktail: FITC-conjugated anti-CD3 (clone SK7) and PE-conjugated anti-CD4 (clone SK3) (5 μL; BD Biosciences) for 30 min at 4 °C. The samples were treated with 400 μL FACS Lysing Solution (BD Biosciences), and after erythrocyte lysis, 10,000 CD3^+^CD4^+^ cells along with beads were acquired on a FACSCanto flow cytometer and analyzed with FACSDiva Software (BD Biosciences) (Figure 1).

The absolute amounts of CD3^+^CD4^+^ lymphocytes in the samples were calculated on a basis of bead and lymphocyte counts. Treg cells (CD4^+^CD25^+^Foxp3^+^) were stained in 200 μL EDTA peripheral blood samples using the Human Regulatory T Cell Staining Kit (eBiosciences, UK), according to manufacturer’s instructions, and acquired on the flow cytometer (clones RPA-T4 and BC96, and for Foxp3 clone PCH101). The absolute counts of Treg were calculated based on the CD4^+^CD25^+^Foxp3^+^ cell percentages of CD3^+^CD4^+^ cells, and the absolute numbers of CD3^+^CD4^+^ cells per microliter.

The statistical analysis was conducted with Statistica 13 software (StatSoft, Poland, Krakow), and the distribution of variables was checked using the Kolmogorow–Smirnoff test. Variables with normal distribution were compared by means of the t-Student test. Categorical variables and variables without normal distribution were compared by means of the U Mann–Whitney test. The cut-off value of Treg count was established by the ROC method. Survival analysis was performed according to the Kaplan Meier method, and log-rank *p* < 0.05 was established as statistically significant.

## 3. Results

### 3.1. The Count of T Regulatory Lymphocytes in the Peripheral Blood of the Study Group

The absolute number of Treg in the peripheral blood did not differ significantly between colon cancer patients (median = 6, IQR = 6) and rectal cancer patients (median = 6, IQR = 9); therefore, these locations were not analysed separately. The number of Treg in the entire study group ranged from 1 to 123 per microliter, with a median value of 6.0 and an IQR of 7. The absolute number of Treg in different subgroups, according to tumor characteristics, was analysed (Table 2).

There was a statistically significant difference between Treg amounts in the Ist and IInd TNM stages (*p* = 0.047), but the differences between stage I and III or IV, or between stage I and pooled stages II-IV, were found not to be statistically significant (*p* = 0.52). The pooled stages I and II had similar Treg amounts to stages III and IV.

There was also no statistically significant difference between the number of Treg in pN- and pN+ patients, as well as between pN0, pN1, and pN2 patients.

The absolute number of Treg cells was higher in pT1 and pT2 tumors than in pT3 or pT4, but without statistical significance. Tumor grading (G) did not influence the number of Treg in the peripheral blood of patients.

### 3.2. The Count of T Regulatory Lymphocytes in the Peripheral Blood of Cancer Patients vs. Control Group

The control group consisted of 21 healthy adult volunteers. The mean absolute number of Treg cells in the control group was 14.9, the median value was 12.5, and IQR was 13.

When compared to the entire study group, the Treg amounts in the control group were found to be higher, with statistical significance (*p* = 0.008) (Figure 2).

There was no statistically significant difference between Treg cell counts in the control group vs. stage I cancer patients. However, again the amount of Treg cells in controls was significantly higher than in patients with stages II and III cancer (*p* = 0.003 and *p* = 0.018).

There was no difference between controls and pT1+pT2 patients, but the pT3+pT4 patients had significantly lower Treg counts in their peripheral blood than the control group (*p* = 0.005).

### 3.3. The Impact of T Regulatory Lymphocytes Count in the Peripheral Blood on Survival

To assess the impact of Treg cells in the blood of the study group on survival, an ROC analysis was performed. The cut-off level was set at 13 cells/μL. The subgroup of patients with up to 12 Treg cells were coded as “Treg-low” and those with 13 or more cells as “Treg-high”.

The Kaplan–Meier plot (Figure 3) showed that in the entire study group, the level of Treg cells in the peripheral blood had no influence on survival.

The analysis in TNM stage subgroups also showed no difference in survival between patients with “low” and “high” Treg counts.

## 4. Discussion

T-regulatory lymphocytes are investigated in cancer patients in several ways. The most common approaches use the identification of Treg cells within peritumoral infiltrate. This provides direct information about the interaction of different cells within the tumor’s microenvironment, and enables their correlation with clinical observations. Another approach investigates the levels of circulating Treg cells in the peripheral blood. Even though the cells in the tumor’s microenvironment are probably more important, circulating Treg cells are most likely involved in the regulation of the immune response to the tumor. As tumor cells spread to other organs through the blood to form metastases, it is possible that Treg cells, which are also circulating in the blood, react with them. While the composition of the peritumoral infiltrate can be analysed only once, the population of Treg cells in the peripheral blood may be examined many times, possibly allowing for cancer monitoring. It would be interesting to assess Treg in both compartments (blood and tumor), but due to limited resources we focused on the blood only.

Our patient population was limited to patients whose tumors were located in the left colon. This is not a weakness of the study, but may be a strength, as our patient population is therefore more homogenous. On the other hand, we cannot simply compare our results with others. Since the method of Treg identification has changed, it is impossible to compare our results with other formerly cited studies. The studies before 2007, and even before 2010, usually addressed T-cells with different characteristics [18,19]. Only the studies that identified Treg as CD3^+^CD4^+^CD25^+^FoxP3^+^ used the same method of cell identification as our study.

We have found that the absolute number of Treg in the peripheral blood of patients with left-sided colorectal cancer was significantly higher in stage I patients than in stage II patients (*p* = 0.047). The difference between stage I and other stages did not reach statistical significance, but there was a tendency for lower Treg counts in stages III and IV than in stage I (Appendix A). This was not investigated in our previous study [9] due to the small number of colorectal cancer patients. Our finding is potentially contrary to results of Sellitto et al. [20], where a linear relationship between the proportion of Treg within CD4^+^ population and the tumor stage was observed. However, our study investigated the absolute count of Treg, which does not always correspond to the mentioned above proportion. The group of patients in the Italian study was only half the size of our patient group. Another study that investigated the quantity of Treg in the peripheral blood of colorectal cancer patients showed that the amount of Treg cells in less advanced stages (I+II) was lower than in more advanced stages (III+IV) [21], but those authors investigated CD4^+^CD25^+^ cells, making their data difficult to compare with the data from our study.

We compared the absolute count of Treg cells in cancer patients and the control group. Our control group was smaller and younger than the studied group. This could influence the results. On the other hand, when we assessed Treg count in patients over 60 years vs. younger, we found no difference (data not shown). The count was significantly lower in cancer patients than in healthy controls. This observation is supported by our study in gastric and colorectal cancer patients [9], where the same trend was observed. However, when only colorectal cancer patients were analyzed, no statistically significant difference was detected. This was most probably due to the small number of colorectal cancer patients involved in that study.

In the present study, when compared to the control group, the count of Treg cells in the peripheral blood was similar to stage I. This finding shows that the early stage of colorectal cancer, at least located on the left side, has no impact on the quantity of T regulatory lymphocytes in the peripheral blood. On the other hand, the accumulation of Treg cells has been described in colonic adenomas, the precancerous lesions [22], and interpreted as the involvement of Treg cells at the very early stage of colonic carcinogenesis. This observation describes the population of Treg within the tissue, not in the peripheral blood, and therefore is not fully contrary to our results. Further analysis showed that there is a statistically significant difference between more advanced, but localized, stages and the control group. When the subgroup of patients who received preoperative radio or radiochemotherapy was excluded, the differences remained statistically significant (data not shown). There was no significant difference between controls and stage IV patients. This could be a surprising observation, but a similar phenomenon was described in the study concerning Treg cells in tumor infiltrating lymphocytes in colorectal cancer patients [23]. In that study, Treg cells were increased in stages II and III, but not in stage IV. Authors speculated that the reason may be the Treg migration from tissue into circulation. Such a hypothesis may also clarify our results. The opposite findings were published in a study of 63 colorectal cancer patients, where the Treg cells were investigated in the blood and tumor tissue [24]. Authors found that there was an increase in the quantity of Treg in the blood and tissue of cancer patients in comparison to healthy controls. However, there were only 2 stage IV patients involved in the study, and their analysis was performed in the pooled I+II and III+IV stages. Their conclusion that there is a correlation of Treg quantity in both compartments with tumor stage may not be fully justified. Additionally, the study of Betts et al. suggested that in colorectal cancer patients in stages II and III the number of Treg in the peripheral blood is higher than in healthy controls [25]. However, they used 35 patients only for the blood testing and 7 healthy volunteers. Moreover, the definition of Treg cells was different as they included CD4^+^CD25^–^Foxp3^+^ cells.

In the final part of the present study, a survival analysis was performed. The ROC analysis method was adopted [26] for establishing the cut-off level. In our patient cohort, we did not find any difference in survival between Treg-high and Treg-low subgroups. We did not find any research directly addressing the issue of circulating Treg cells and survival. There are contradictory data concerning the prognostic value of the density of Treg cells within tumor tissue. A meta-analysis published in 2015, which included eight studies on colorectal cancer patients [27], concluded that the presence of Treg within TIL is a positive prognostic factor.

All mentioned above controversies indicate that there is a need for further, well designed studies with sufficient numbers of patients in subgroups, analyzing Treg importance in various compartments. This may help in improving recently accepted immunotherapy strategies.

## 5. Conclusions

In summary, the results from the current study show that the absolute number of T-regulatory lymphocytes in the peripheral blood of patients with left-sided colorectal cancer was significantly decreased in comparison to healthy controls. When the study group was split into subgroups, the decrease was only found to be significant between the control group and stage II and III patients. In stage I, when the disease is highly localized, the peripheral levels of Treg do not differ much from those in healthy controls. In stages II and III, as the disease progresses, the Treg levels drop significantly in the periphery. In stage IV, the disease becomes systemic, and Treg cell redistribution between compartments occurs. The presence of CD4^+^CD25^+^FoxP3^+^ lymphocytes in the peripheral blood had no impact on survival.

## Figures and Tables

**Figure 1 medicina-55-00307-f001:**
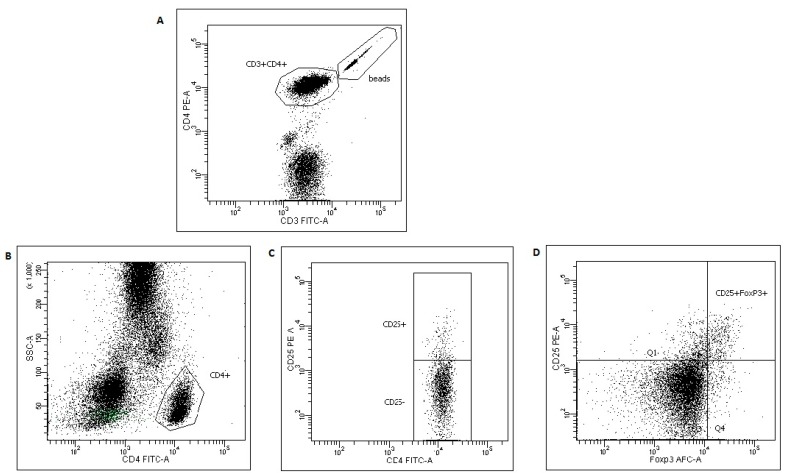
Gating strategy and analysis of Treg lymhocytes. Absolute numbers of CD3^+^CD4^+^ lymphocytes were calculated on a basis of bead numbers (**A**). During Treg cells analysis, CD4^+^ lymphocytes (**B**) were divided into CD25^+^ and CD25^-^cells (**C**), and then Treg cells were determined as CD25^+^FoxP3^+^ lymphocytes (**D**).

**Figure 2 medicina-55-00307-f002:**
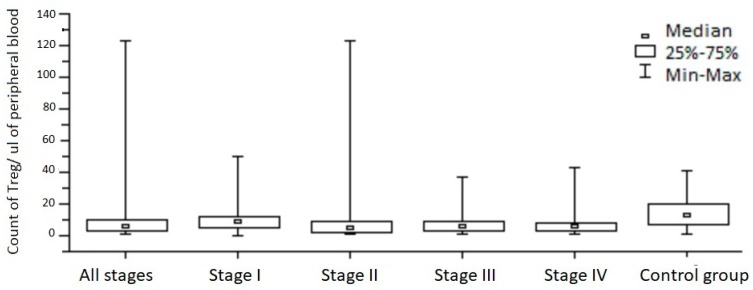
The absolute number of Treg lymphocytes (CD4^+^CD25^+^FoxP3^+^) in the peripheral blood of patients with left-sided colorectal cancer and the control group.

**Figure 3 medicina-55-00307-f003:**
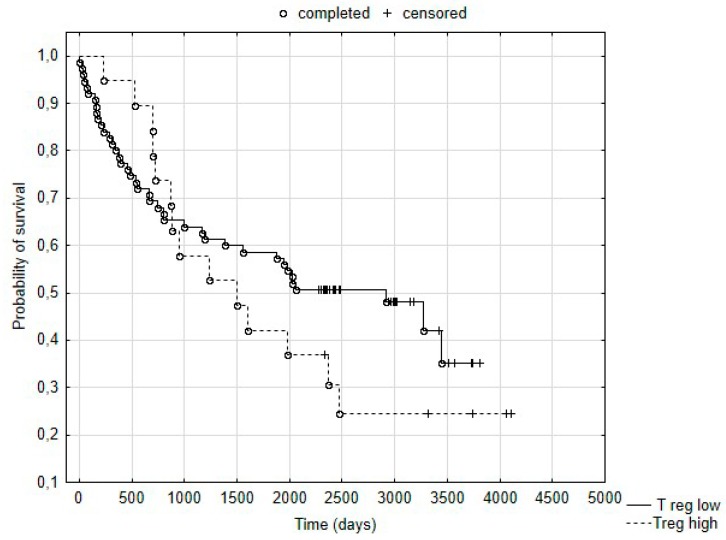
Probability of survival in “Treg-low” and “Treg-high” groups of left-sided colorectal cancer patients.

**Table 1 medicina-55-00307-t001:** Clinicopathological characteristics of the patients.

Tumor Location	Number of Patients
Left colon	43
Rectum	51
T1	3
T2	15
T3	58
T4	18
N0	42
N1	28
N2	20
Nx	4
M0	73
M1	21
Stage I	15
Stage II	28
Stage III	30
Stage IV	21
Grade 1	27
Grade 2	51
Grade 3	10
Grade not assessed	6
R0	69
R1	3
R2	22

**Table 2 medicina-55-00307-t002:** Characteristics of Treg in the peripheral blood of the study group.

Group	Mean	Median	Min	Max	IQR
Study group	10.35	6.0	1	123	7
Stage I	14.46	9	1	52	8
Stage II	6.82	5	1	123	7.5
Stage III	8.9	6.5	1	37	6
Stage IV	9.4	6	1	43	5
pN0	9.5	5.5	1	52	8
pN1	9.96	6	1	37	9
pN2	7.75	5.5	1	43	5.5
pT1	11.33	12	9	13	4
pT2	12.7	7	1	52	8
pT3	7.8	6	1	37	6
pT4	10.6	6.5	2	123	14
G1	8.5	5	1	45	8
G2	10.2	7	1	123	8
G3	7.7	5	1	21	7

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
