# Peer review of "T Regulatory CD4+CD25+FoxP3+ Lymphocytes in the Peripheral Blood of Left-Sided Colorectal Cancer Patients"

_medicina, 2019, doi:10.3390/medicina55060307_

Round 1

Reviewer 1 Report

I have read with great pleasure this article: hghly focused and therefore very clear, and also very well documented.

Plus: 

well identified Treg population (CD3+CD4+CD25+FoxP3+) and assessment of absolute Treg values in blood;

well characterized cohort of patients with left-sided colon cancer (43) and rectum cancern (51) (this is an advantage for the cohort uniformity, but it is limiting the relevance of the study);

investigation of a blood biomarker which allows monitoring of patients;

interesting observation of the particular Tregs status in stage II as compared to stage I, that may be used for monitoring disease progression in early stages;

clear presentation and comparison of the obtained data with other studies;

good English.

Minus:

the group of controls included to few subjects (21). Thish may affect the validity of the comparison with the patients' group (94) taking into account inter-individal variabilility. Moreover, the patients and control groups were not age-matched (controls were significantly younger than patients) and this may impact results as blood Tregs number may vary due to age. Nevertheless, it is expected that Tregs increase with age (PMID: 28253320)) and therefore the control cohort used in the present study does not change the conclusions (left-sided colon+rectum cancer patients have lower number of Tregs in blood as compared to the control group which included younger subjects than the investigated patients).

the study was exclusively focused on blood Tregs and did not provide correlative information regarding Tregs in the (peri)tumoral region. This correlation would have been highly interesting for elucidating the trafick of Tregs between blood and tumor in various stages of disease.

moreover, investigations on a group of patients with left-sided colon cancer would have clarified if Treg status the blood is more or less correlated with tumor localization. Nevertheless, the study brings valuable data on the Tregs status in the blood of left-sided colon cancer and rectum cancer patients, and this is a step forward in the field.

I suggest to mention these aspects in the manuscript (part were already mentioned) when describing the limitations of the study.

Some minor typing errors are still in the manuscript (page 7).

Author Response

Response to Reviewer 1 Comments

Dear Reviewer,  we are grateful for your constructive review and comments.

The comment concerning the number and the age of the control group is important. Therefore we added in the discussion section following information: “Our control group was smaller and younger than the studied group.  This could influence the results. On the other hand when we assessed  Treg count in patients over 60 years  vs younger, we found no difference (data not shown). “  When we divided the CRC patients group in <=60 and  >60 there was 31 patients in the younger and 63 in the older subgroups.  There was no difference between these groups. It was impossible to increase the number of patients in the control group because this study finished.

It would be interesting  to extend the  research to the  Treg in the tissues as well as the involvement of  right sided tumors.  We changed the discussion to point this out. Unfortunately we had no resources  to perform such an extended study.

There are some additional corrections in the text resulted from the comments of  the second Reviewer as well as the mistakes we have found reading the text again.

Reviewer 2 Report

In the present manuscript entitled “T regulatory CD4+CD25+Foxp3+ lymphocytes in the peripheral blood of left sided colorectal cancer patients”, Dylag-Trojanowska and colleagues evaluated the circulating regulatory T cell (Treg) compartment of 94 left sided colorectal cancer (CRC) patients as well as its prognostic significance. The authors report that circulating Treg numbers are significantly decreased in left sided CRC patients in comparison to healthy controls and dismiss the possibility Treg numbers are a biomarker of CRC patient long-term survival.

Although this is an interesting study, involving a quite reasonable number of CRC patients, there are some limitations:

1.   The finding that CRC patients have lower number of circulating Tregs in comparison to healthy controls is somehow unexpected given the available literature on CRC and should be discussed. The observation that Treg numbers tend to decrease as tumor stage increases is, as acknowledged by the authors, controversial. This important issue should be better discussed and additional relevant publications included in this discussion (e.g.  Betts G et al. Gut. 2012; 61:1163-71).

2.   How is the frequency of circulating Tregs in patients at the different disease stages and in comparison with healthy controls? Can representative FACS plots be shown?

3.   The authors should clarify in the Material and Methods section at which time the blood collection for patient analysis was performed as well as indicate how was the gate strategy uniformized if analyses were performed in different days in order to allow for comparison?

The antibody clones used in flow cytometry analysis should also be indicated in this section.

4.   The authors should explain why they opted to include some rectal cancer patients that were exposed to preoperative chemotherapy and chemoradiotherapy in the analyses. Is there any possibility that the preoperative treatment impacted on their Treg numbers? How are the results if these patients were to be excluded?

5.   The following sentence: ”However, our study investigated the percentage of Treg within CD4+ cells, …” (pag. 7, lines 186-187) is unclear and should be clarified or otherwise corrected in the text.

Author Response

Response to Reviewer 2 Comments

Dear Reviewer,  we are grateful for your constructive review and comments.

We agree, that there are some studies which could be included in the disscussion. According to the suggestion we analysed the findings in the paper of Betts et al. This is included into disscussion section.  The data concerning the number and the distribution of Treg cells in various stages and subgroups are presented in table 2.  Similar data for the control group was included in  the text. 

The section Material and methods has been extended and all the suggested issues were included. This section is also enriched with the FACS plot showing  the gating strategy.

In our study we included 6 rectal cancer patients with  preoperative radiotherapy or radiochemotherapy. There was the mistake in the previous version of the text where preoperative chemotherapy was used instead of radiotherapy . This was amended. The reason why we have not excluded this group of patients was, that it became a standard therapy in locally advanced rectal cancer.  Of our patients with preoperative treatment four  were in stage II and two in stage III. 

When excluded from analysis  there was still the statistically significant difference in Treg numbers between  all CRC patients and controls (p=0.01)  as well as between stage II vs controls and stage III vs controls (p=0.004 and p=0.02 respectively). The comment has been made in the discussion section.

The sentence about the study of Sellito et al has been amended on page 7.

It would be interesting  to extend the  research to the  Treg in the tissues as well as the involvement of  right sided tumors.  We changed the discussion to point this out. Unfortunately we had no resources  to perform such an extended study.

We are sending to the Editor  the plots you requested for but in our opinion they will add a little value to the manuscript. ( Fig 4. Exemplary plots of patients from  the control group (A) , stage I (B) , stage II (C), stage III (D) and stage IV (E)) – it can be a supplementary material to the manuscript.

There are some additional corrections in the text resulted from the comments of  the second Reviewer.

Round 2

Reviewer 2 Report

In the manuscript entitled “T regulatory CD4+CD25+Foxp3+ lymphocytes in the peripheral blood of left sided colorectal cancer patients”, Dylag-Trojanowska et al analyzed the circulating regulatory T cell (Treg) compartment of 94 colorectal cancer (CRC) patients as well as its prognostic significance. The authors show that circulating Treg numbers are significantly decreased in CRC patients in comparison to healthy controls and demonstrate that Treg numbers are of no prognostic value for CRC patient’s long-term survival.

This is an interesting study and I am satisfied with the revised version the authors provided. I therefore recommend this revised manuscript for publication at Medicina.